# Protective Effect of Ethoxyquin and N-acetylcysteine on Biochemical and Pathological Changes Induced by Chronic Exposure to Aflatoxins in Laying Hens

**DOI:** 10.3390/toxins17100514

**Published:** 2025-10-20

**Authors:** María Carolina de-Luna-López, Arturo Gerardo Valdivia-Flores, Teódulo Quezada-Tristán, Raúl Ortiz-Martínez, Erika Janet Rangel-Muñoz, Emmanuel Hernández-Valdivia, Esther Albarrán-Rodríguez, Elizabeth de Santiago-Díaz

**Affiliations:** 1Departamento de Ciencias Veterinarias, Centro de Ciencias Agropecuarias, Universidad Autónoma de Aguascalientes, Aguascalientes 20131, Mexico; carolina.deluna@edu.uaa.mx (M.C.d.-L.-L.); teodulo.quezada@edu.uaa.mx (T.Q.-T.); raul.ortiz@edu.uaa.mx (R.O.-M.); janet.rangel@edu.uaa.mx (E.J.R.-M.); emmanuel.hernandez@edu.uaa.mx (E.H.-V.); elizabeth_8552@hotmail.com (E.d.S.-D.); 2Departamento de Medicina Veterinaria, Centro Universitario de Ciencias Biológicas Agrícolas, Universidad de Guadalajara, Guadalajara, Jalisco 45110, Mexico; esther.albarran@academicos.udg.mx

**Keywords:** aflatoxicosis, glutathione S-transferase, reduced glutathione, liver pathology, poultry

## Abstract

Aflatoxins (AFs) represent a major threat to poultry health and food safety due to their hepatotoxic, immunosuppressive, and carcinogenic effects. This study evaluated the chemoprotective potential of ethoxyquin (EQ) and N-acetylcysteine (NAC) in laying hens (80.8 and 33.3 mg/kg BW/d) exposed to chronic dietary AFs contamination (0.0–1.5 mg/kg). A total of 360 Hy-Line W36 Leghorn hens were monitored over 72 weeks using biochemical and histopathological analyses of liver and kidney tissues. NAC significantly (*p* < 0.01) increased hepatic and renal levels of reduced glutathione (GSH) and stimulated glutathione S-transferases (GST) and gamma-glutamyl transferase (GGT) activity, enhancing detoxification. Both agents significantly (*p* < 0.05) reduced plasma ALT and AST levels, preserved total protein concentrations, and attenuated liver and kidney hypertrophy. EQ demonstrated antioxidant effects, stabilizing enzymatic responses and limiting tissue damage. Histopathological analysis revealed fewer structural alterations and cellular degeneration, especially in the NAC-treated group (*p* < 0.01). These results suggest that NAC and EQ activate endogenous detoxification mechanisms, both enzymatic and non-enzymatic, effectively mitigating chronic aflatoxin toxicity. Their dietary supplementation offers a safe and sustainable chemoprotection strategy to support poultry health and productivity, particularly in regions facing high mycotoxin exposure.

## 1. Introduction

Mycotoxins are secondary metabolites produced by filamentous fungi that contaminate crops before or after harvest, compromising food safety and affecting human and animal health. In the poultry industry, aflatoxins (AFs), ochratoxins, trichothecenes, and citrinin stand out due to their toxicological relevance. Furthermore, AFs are the most studied mycotoxins due to their high toxicity, carcinogenic potential, and persistence in food products [1]. The AFs are a major risk because they are biotransformed in the liver into reactive metabolites, called AFs-epoxides (AFBO), that cause hepatic, renal, and immunological damage [2]. In laying hens, chronic exposure to low or moderate levels of AFs is associated with immunosuppression, toxic residues in edible products, reduced feed efficiency, and declines in egg production and quality, as well as toxic residues in edible products [3,4,5]. In response to AFs exposure, the hen organism activates endogenous detoxification pathways to mitigate the negative impact of aflatoxins, including glutathione S-transferases (GST) and other hepatic detoxification systems. Notably, reduced glutathione (GSH) plays a crucial role as a non-enzymatic defense, directly neutralizing AFs-free radicals and participating in the conjugation of toxic compounds. This action is complemented by GST, which catalyzes the binding of GSH to reactive metabolites, facilitating their elimination. The synergy between GSH and GST is fundamental to the cellular response against oxidative stress and AFs-induced toxicity [6,7].

Poultry production is progressively expanding in all producing countries, supported by population growth and sustained demand. However, the poultry industry is challenged by unprecedented issues such as rising input costs, prevalent and emerging diseases affecting productivity, as well as complex marketing systems and trade barriers [8,9,10]. Despite advances in the management of mycotoxins in poultry production, there remains a need to strengthen endogenous defense mechanisms, particularly against chronic and acute exposure to AFs. In this context, chemoprotection strategies have emerged, defined as the use of natural or synthetic agents to reduce cancer risk or delay its onset or recurrence. Additionally, in the agri-food sector, the chemoprotection strategy aims to mitigate the effects of toxic compounds like AFs. Due to their high sensitivity, birds have become key experimental models in the discovery of novel chemopreventive agents against AFs [11]. Stimulation of the defense mechanisms, bioactive agents such as ethoxyquin (EQ) and N-acetylcysteine (NAC) have been used, showing efficacy in animal models. NAC acts as a precursor of GSH, enhancing its endogenous synthesis and reinforcing the organism’s antioxidant capacity. This increase in GSH not only enables more effective neutralization of free radicals but also boosts GST activity in the conjugation of toxic metabolites derived from AFs. On the other hand, EQ, on the other hand, is widely used in animal feed due to its antioxidant and lipid-stabilizing properties [12,13,14,15,16].

Studying these compounds as chemoprotective strategies in laying hens chronically exposed to aflatoxins is essential to safeguard animal health, enhance productivity, and ensure the safety of poultry products. Moreover, it contributes to the development of sustainable nutritional solutions aligned with the One Health approach [16].

Based on the above, the objective of this study was to evaluate the effect of two chemoprotective agents, ethoxyquin and N-acetylcysteine, administered through diet, on the biochemical and pathological parameters resulting from chronic aflatoxin exposure in laying hens. Additionally, the present study aimed to stimulate endogenous detoxification mechanisms against aflatoxins, encompassing both enzymatic systems and non-enzymatic systems, to identify the ability of chemoprotective agents in mitigating AFs-induced hepatic and renal damage and provide scientific evidence for the design of sustainable feeding strategies that promote animal health and food safety.

## 2. Results

### 2.1. Animal Performance

None of the hens included in this study showed any abnormalities before the observation period, meaning that their development was normal for their age. All the hens in the control, NAC, and EQ groups remained healthy throughout the 72-week observation period. However, birds exposed to aflatoxin-contaminated feed showed signs of depression, immobility, rough plumage, nasal discharge, and decreased body condition during the first two weeks, but no mortality attributable to the intoxication. The productive performance of the hens showed small but statistically significant differences in feed intake, daily egg production per bird, egg weight, and, therefore, in the feed efficiency (Table 1). These variations were observed from the start of laying at 14 weeks of age and continued until the end of the three laying periods (72 weeks). No evidence was found that the intake of NAC or EQ altered the health of the hens.

### 2.2. Biochemical Analysis

#### 2.2.1. Reduced Glutathione

Hens exposed to AFs showed a decrease in hepatic GSH concentration, especially at doses of 1.0 and 1.5 mg/kg after 72 weeks of intoxication (Figure 1: Table 2). NAC significantly increased (*p* < 0.01) GSH levels in intoxicated hens and in those that received only the chemoprotectant between weeks 14 and 46; however, GSH depletion occurred afterward, possibly due to GSH consumption in AFs metabolism. Although NAC maintained stable levels in the control group, its concentration declined when combined with AFs. In contrast, hens treated with EQ exhibited a smaller increase in GSH, suggesting an antioxidant mechanism rather than direct stimulation of its synthesis, as well as less variation in hepatic tissue concentration, which could indicate lower AFBO production and reduced reliance on the mercapturic acid metabolic pathway. In renal tissue, all groups experienced a decrease in GSH at 72 weeks, but NAC provided greater protection by preserving levels compared to the AF group. Although EQ treatment behaved similarly to the control and NAC groups, its protective effect did not surpass NAC, despite significant differences (*p* < 0.01) observed at 46 and 72 weeks. No significant differences were observed between the biochemical values of the hens that ingested NAC or EQ, and the biochemical levels observed in the liver, kidneys, and plasma of the control group of hens that did not consume AFs or chemoprotectors.

#### 2.2.2. Glutathione S-Transferases

The specific enzyme activity of glutathione S-transferases showed a characteristic pattern within the 72-week observation period; this pattern of enzyme activity showed an initial increase followed by a stabilization phase and increased reactivity at the end of the period (Figure 2). From week 20 of observation, the effects of AFs and chemoprotectants on GST activity stabilized, with a significant decrease (*p* < 0.01) observed at 46 and 72 weeks of intoxication. GST activity in the AFs group decreased significantly (*p* < 0.01) at doses of 1.0 and 1.5 mg/kg, likely due to GSH depletion in the liver caused by AFs metabolism. The reduction in GST was more pronounced in the AFs group compared to those treated with chemoprotectants (Figure 2), suggesting an increase in the protective activity of the latter. In contrast, the increase in GST activity between weeks 46 and 72 coincided with hepatic GSH depletion (Figure 1). In renal tissue, GST activity stabilized from week 20, but all AFs treatments showed a significant increase (*p* < 0.01) compared to the control, particularly at 46 and 72 weeks. Unlike in the liver, enzymatic activity in the kidney remained elevated, possibly due to lower tissue damage. The NAC group exhibited a greater induction of GST activity compared to AFs and EQ (Figure 2), indicating its possible influence on GSH regulation. Finally, the increase in GST between weeks 46 and 72 coincided with GSH depletion during the same period (Figure 1).

#### 2.2.3. Gamma Glutamyltransferase

The activity of GGT in the liver increased significantly (*p* < 0.01) in hens exposed to 1.0 and 1.5 mg/kg of AFs at 20 weeks, in response to the presence of AFBO-GSH conjugates. Over time, this activity decreased (*p* < 0.01) until the end of this study due to its role in the elimination of conjugates via the mercapturic acid pathway. In the NAC group, GGT increased during the first 20 weeks but did not show sustained induction. In contrast, EQ exhibited more stable activity, possibly by reducing AFBO formation and limiting the use of the mercapturic acid pathway. In renal tissue, GGT activity was lower and more stable in the AFs, NAC, and EQ groups, with significant variations (*p* < 0.01) during the first 20 weeks. NAC showed fluctuations but failed to maintain prolonged induction, whereas EQ exhibited an increase proportional to the AFs dose, suggesting a chronic effect at the end of the experiment (Figure 3).

#### 2.2.4. Alanine Aminotransferase and Aspartate Aminotransferase

Hens intoxicated with AFs exhibited a significant increase in plasma ALT activity, indicating cellular damage, with the greatest effect observed at week 72. NAC protected by preventing ALT elevation until week 46, though an increase (*p* < 0.05) was recorded in the final period. EQ also showed a protective effect in weeks 20 and 46, but enzymatic activity increased in week 72 (Figure 4).

AST activity also increased (*p* < 0.01) in the AFs group, especially at doses of 1.0 and 1.5 mg/kg, reflecting cellular destruction. However, in 46 and 72 weeks, AST activity decreased (*p* < 0.01) compared to the control (0.0 mg AFs/kg), suggesting a resistance mechanism in chronic intoxication. During the first 20 weeks, EQ elevated AST activity, but in the latter half of this study, it successfully reduced it. In contrast, NAC did not preserve cellular integrity, as AST activity remained elevated (*p* < 0.01) throughout the entire experimental phase (Figure 4).

#### 2.2.5. Total Proteins

In the liver, protein concentration increased significantly (*p* < 0.01) at 72 weeks in intoxicated hens, with levels surpassing those of the control group (0.0 mg AFs/kg). This trend was also observed in hens treated with NAC and EQ alongside AFs, suggesting functional recovery related to GSH metabolism. In the kidney, protein concentration in AFs-intoxicated hens increased over time, although it remained significantly reduced (*p* < 0.01) compared to the control (0.0 mg AFs/kg). In contrast, in the NAC and EQ groups, renal protein levels remained stable in the second half of this study, indicating a possible protective effect. Hens exposed to AFs showed a significant decrease (*p* < 0.01) in plasma proteins until week 46, followed by stabilization. In groups with chemoprotectants, the reduction was less severe, particularly with NAC and EQ (Figure 5).

### 2.3. Morphological Changes

#### 2.3.1. Relative Weight of Liver and Kidneys

Hens exposed to AFs at doses of 1.0 and 1.5 mg/kg showed a significant increase (*p* < 0.01) in relative liver weight from week 46 onward, an effect associated with AFs concentration and treatment duration. NAC protected against AFs until week 46; however, in week 72, an increase in liver weight (*p* < 0.05) was recorded, indicating a reduction in its protective effect. In the groups treated with EQ + AFs (0.5, 1.0, and 1.5 mg/kg), hepatic enlargement was also observed (*p* < 0.05), whereas hens that received only EQ without AFs showed no changes, suggesting a protective action. Regarding renal weight, hens exposed solely to AFs showed no changes (*p* > 0.05); however, the groups with NAC + AFs and EQ + AFs exhibited increases in weeks 46 and 72, possibly due to a toxic effect of the chemoprotectants or an additive interaction with AFs (Figure 6).

#### 2.3.2. Histopathology

Microscopic analysis of the hepatic and renal tissues of hens in the control group (0.0 mg of AFs/kg of feed), the group treated with 0.5 mg of AFs/kg of feed, as well as the groups that received NAC and EQ without aflatoxin exposure revealed no morphological alterations attributable to aflatoxin toxicity at the 0.5 mg/kg dose or to chemoprotective agents. As in the control group, hens that ingested NAC or EQ but did not consume AFs showed a characteristic structural arrangement of the tissues, and no macroscopic or histological lesions were observed in their liver or kidneys. In contrast, hens exposed to 1.0 and 1.5 mg of AFs/kg of feed without chemoprotective treatment showed microscopic alterations in liver and kidney tissues, apparently induced by the ingestion of the mycotoxin (Table 3 and Table 4).

Hens intoxicated with 1.0 and 1.5 mg of AFs/kg of feed exhibited microscopically evident hepatic alterations with statistically significant differences (*p* < 0.05). The main lesions included lymphocytic hepatitis, characterized by foci of inflammatory lymphocyte infiltration located around blood vessels and within the liver’s perisinusoidal space; cellular hydropic degeneration, evidenced by excessive intracellular water accumulation leading to cellular swelling, potentially progressing to necrosis; and steatosis, identified by visible lipid accumulation as cytoplasmic droplets within hepatocytes (microvesicular type observed at 14 and 20 weeks of intoxication, and macrovesicular type at 46 and 72 weeks) (Figure 7). Hemorrhagic foci, necrotic areas, and loss of structural organization were also observed. In contrast, hens exposed to the same AFs doses but treated with NAC or EQ showed a significant reduction (*p* < 0.05) in these morphological alterations.

As observed in the liver, the kidneys of hens intoxicated with 1.0 and 1.5 mg of AFs/kg of feed exhibited a significantly higher number of microscopic morphological alterations (*p* < 0.05) compared to the control groups and those that, although intoxicated, received chemoprotective treatment (Table 4). The main alteration identified was lymphocytic nephritis, characterized by basophilic cell aggregates located at the periphery of blood vessels and between glomeruli and renal tubules. Additionally, ureteritis, congestion, thrombosis, and hemorrhages were observed (Figure 8), along with lesser occurrences of glomerular proliferation, glomerulonephritis, tubular degeneration, detachment of the brush border in renal tubules, and necrotic foci. In hens intoxicated with 1.0 and 1.5 mg of AFs/kg of feed that received chemoprotectors, a significant reduction (*p* < 0.05) in these renal alterations was observed, particularly in those treated with 1.5 mg of AFs/kg combined with EQ.

## 3. Discussion

This study analyzed that chronic exposure to aflatoxins in the feed of Leghorn laying hens caused the development of a pattern of clinical manifestations with biochemical changes and permanent pathological lesions in the liver and kidneys. Several of our results are consistent with previous studies on aflatoxicosis in adult hens [5,17]. This study provides evidence that ingestion of ethoxyquin and N-acetylcysteine can mitigate the alterations caused by chronic exposure to aflatoxins without affecting the health of laying hens. To our knowledge, many of these alterations in plasma and cellular biochemistry and histology, as well as the chemoprotection provided by these compounds, are published for the first time in this study. These findings are relevant for making an accurate diagnosis and preventing chronic aflatoxicosis in laying hens, as well as for designing appropriate measures to protect public health and the poultry industry.

As in other studies [18] on natural mycotoxin contamination in hen feed, low concentrations of ZEN, OTA, DON, and FB were found. The presence of FB, DON, and OTA was below the minimum detection limits, while ZEN concentration was only detected occasionally during weeks 14 and 46 of this study (63.1 ± 24.4 μg/kg). The concentration of ZEN in feed that could cause a decrease in bird performance is >50 mg/kg [19]; however, in this study, the observed ZEN concentration was several times lower than that concentration, suggesting that ZEN or other mycotoxins did not cause the health changes detected in this study. Liquid chromatography-tandem mass spectrometry is widely used as the reference method for the routine determination of mycotoxins. The main advantages of this method are its low detection and quantification limits, as well as its applicability to different feed varieties, as it is a simple, fast method suitable for the simultaneous determination of multiple mycotoxins [20]. In this study, the concentration of aflatoxins was determined by HPLC, while the absence of other mycotoxins was verified by an indirect enzyme-linked immunosorbent assay, following extraction, purification in immunoaffinity columns, and comparison with the results of purified mycotoxin standards. This ELISA procedure for the determination of mycotoxins is acceptable to several authors [21,22], who agree that there is a very high correlation (R^2^ > 90%) between the concentrations of mycotoxins in feed and animal products, estimated comparatively between ELISA and LC-MS/MS. Therefore, it can be assumed that both techniques are satisfactory in terms of meeting feed safety control basic requirements.

In several countries, data on poultry feed contamination is limited; however, studies indicate that AFs contamination in feed is common [23], making chronic exposure in vulnerable production systems a plausible concern. Previous studies [17] have shown that normal homeostasis in laying hens continues to function regularly, even at doses as high as 1.5 mg of AFs per kg of body weight. Therefore, the high levels of aflatoxins used in this study were selected to challenge the efficacy of chemoprotectors, even in situations of high exposure in regions with limited quality control measures. The concentrations of NAC and EQ used for dietary supplementation were selected based on efficacy and safety criteria reported in the literature. For NAC, a concentration of 80.8 mg/kg BW/d was used; this dose has been previously employed in studies with broiler chickens to assess its protective properties [12] and falls within the clinically recommended range for humans (600–1500 mg/day) as a mucolytic and antioxidant agent [24]. Diverse studies have evaluated the main biomarkers derived from the therapeutic administration of NAC to validate its bioavailability and efficacy, especially cysteine, glutathione, and the enzymes involved in its binding to xenobiotic compounds [12,13,25,26]. In this study, the concentration of GSH in liver and kidney tissues was measured, as well as the enzymatic activity of glutathione S-transferases; however, no distinction was made between compounds derived from the ingestion and metabolism of NAC and the *de novo* synthesis of GSH or the enzymatic induction of GST, so their concentration or activity cannot be considered a direct biomarker of the chemoprotective efficacy of NAC treatment. Nevertheless, the biological effects of NAC treatment were evaluated through functional, biochemical, and histopathological parameters in birds intoxicated with AFs.

Exposure to AFs significantly reduced hepatic GSH levels, as the liver is the primary target organ and contains higher baseline GSH concentrations than the kidneys [27,28]. In addition, the liver is responsible for synthesizing and releasing more than 90% of the total GSH present in plasma and serves as a key organ in detoxification mechanisms. In parallel, the kidney facilitates GSH catabolism through the activity of the GGT enzyme [29,30]. The hepatic depletion observed in this study aligns with previous findings in acutely intoxicated broilers [4,31]. Similarly, renal GSH concentration was significantly reduced in AFs-treated hens, likely due to inherently lower GSH levels in the kidneys and limited capacity to form and eliminate AFs-conjugates via the mercapturic acid pathway. Reduced glutathione depletion has been associated with cellular damage and organ dysfunction in response to toxic insults [6,29]. N-acetylcysteine, on the other hand, exerted a protective effect by elevating GSH concentrations in both organs, likely due to its role as a cysteine donor and its ability to competitively bind reactive AFs metabolites [2,12,32]. In contrast, ethoxyquin (EQ) did not show protective effects on GSH concentrations in either the liver or the kidney. While EQ has been reported to induce hepatic GSH conjugation activity in rats [33], limited studies in poultry have failed to demonstrate significant benefits [34,35].

Glutathione transferases are key isoenzymes in detoxification, responsible for catalyzing the conjugation of GSH with metabolites of toxic compounds such as aflatoxins [36]. Their activity, stimulated by adequate intracellular GSH levels, contributes to interspecies differences in AFs resistance [6,11,27]. In this study, hepatic GST activity decreased in hens treated with 1.0 and 1.5 mg/kg of AFs toward the final phase, whereas renal activity increased across all AFs levels throughout the exposure period. Treatment with NAC promoted GST activity in both liver and kidney, consistent with findings in rats and poultry [12,25]. Ethoxyquin, known to induce hepatic and renal detoxifying enzymes such as GST and CYP450 in several species [14,37], significantly increased hepatic GST activity in intoxicated hens. In contrast, in kidney tissue, EQ seemed to exert mainly antioxidant effects, rather than enzymatic induction. Current evidence on its protective role in poultry remains limited and is primarily based on studies in other animal models [37,38,39].

Gamma-glutamyltransferase activity increases in response to pathological processes affecting organs such as the liver and kidneys. Cellular damage induced by aflatoxins can lead to membrane rupture and the release of GGT into the bloodstream, making plasma GGT a useful biomarker [30,40]. Hepatic GGT elevation is typically associated with inflammation, biliary stasis, or bile duct hyperplasia [41,42]. In this study, hens exposed to AFs exhibited decreased hepatic GGT activity toward the end of the experiment, while renal activity increased transiently during the early phase. Although mammalian kidneys are known for their high GGT activity and their role in the elimination of xenobiotics [43,44], studies in poultry suggest greater GGT activity in the liver [45,46]. Previous studies in AFs-intoxicated broilers and laying hens have reported elevated plasma GGT levels, indicating liver and kidney injury [47,48]. However, other reports have shown hepatic GGT suppression with similar AFs doses, and some have found no significant changes [5,49]. N-acetylcysteine increased hepatic GGT activity during the first 20 weeks of treatment, with no significant effect on the kidney. While poultry-specific data is lacking, studies in mammals suggest NAC may regulate enzymes involved in GSH homeostasis and protect renal function against xenobiotics [13,26]. Ethoxyquin did not increase hepatic GGT activity in intoxicated hens but appeared to reduce the formation of toxic metabolites in renal tissue, resulting in low enzyme activity. As with NAC, evidence in poultry is limited, though EQ has been reported to induce hepatic GGT in AFB_1_-intoxicated rats to facilitate conjugate excretion via the mercapturic acid pathway [50].

Most plasma proteins are synthesized in the liver, and their concentrations can be altered by pathological processes that affect protein synthesis or plasma water balance [51]. In birds, exposure to aflatoxins reduces total protein levels in the liver and kidneys due to the breakdown of polyribosomes and degranulation of the rough endoplasmic reticulum, which disrupts RNA synthesis and, consequently, protein production [52]. In this study, hens exposed to AFs for 72 weeks showed decreased total protein concentrations in plasma and kidney tissue, reflecting hepatic and renal damage caused by the mycotoxin. These findings are consistent with previous reports on birds treated with various AFs doses [3,31,48]. Administration of NAC mitigated protein loss in plasma and liver, suggesting a protective effect on organ function. Similar improvements were reported in AFs-intoxicated birds treated with NAC [12,13]. Likewise, EQ improved total protein levels in plasma, liver, and kidney. However, specific data on EQ’s effects against AFs toxicity in laying hens are still lacking.

Alanine aminotransferase and aspartate aminotransferase are present in high concentrations in the liver and kidneys, and their release into the bloodstream due to tissue necrosis makes them sensitive biomarkers of hepatic and renal damage [53]. In this study, hens exposed to aflatoxins showed increased plasma activity of ALT toward the end of the experiment, whereas AST activity rose during the early stages. These changes reflect tissue injury caused by AFs and are consistent with previous studies linking elevated ALT to liver and kidney damage, and decreased AST activity to protein synthesis inhibition [1,3]. Treatment with NAC offered protection against increased ALT but did not alter AST levels. Similar results have been reported in AF-intoxicated chickens, where NAC improved ALT activity without affecting AST [54]. In rats, NAC alone increased ALT activity without histopathological evidence of organ damage [55]. Ethoxyquin also reduced plasma ALT and AST activity throughout the experiment, possibly due to its antioxidant properties [56]. However, clear evidence on its mechanisms against AFs toxicity in poultry remains limited.

The applied treatments caused variations in relative liver weight, the primary target organ affected by aflatoxins. In this study, liver weight significantly increased with 1.0 and 1.5 mg/kg AFs over 72 weeks, likely due to lipid accumulation. This aligns with previous reports on birds exposed to similar or higher AFs levels over short or acute periods [4,40,57]. Chronic aflatoxin exposure has been associated with hepatomegaly, as increased lipid accumulation, together with the inhibition of hepatic lipid transport, contributes to liver mass enlargement [58]. In contrast, kidney weight remained unaffected, even at the highest aflatoxin doses, consistent with findings from [59]. This suggests kidneys may be less sensitive to chronic AFs toxicity. Administration of NAC partially attenuated the liver weight increase, particularly up to week 46. However, evidence on NAC’s protective effects in birds remains limited and inconsistent [12]. Conversely, EQ did not mitigate hepatic toxicity and was associated with increased kidney weight in birds treated with 0.5 and 1.0 mg/kg AFs. While EQ has been studied as a modulator of detoxification pathways in other species, its role in birds exposed to aflatoxins remains poorly documented [14,34].

In this study, hematoxylin and eosin staining was used to detect fatty liver degeneration; these tissue changes were detected with sufficient accuracy to identify them as hepatic steatosis. This staining has also been described in other studies with birds and mammals [40,42,47] and has been linked to alterations in lipid metabolism enzymes [28,49]. Other specific stains, such as Oil Red O or PAS with diastase, are highly efficient and enable the distinction between hepatic steatosis and lipid accumulation, glycogen vacuolization, or hydropic changes [60]; however, H&E staining remains a widely accepted method for identifying both microvesicular and macrovesicular hepatic steatosis [61]. H&E staining has also been reported as a method for identifying lipid accumulation in the liver, provided that specific morphological criteria are considered. These criteria include the presence of intracytoplasmic vacuoles, peripheral displacement of the nucleus, the appearance of the vacuoles, and the characteristic zonal distribution [12,23,42,46]. Lipid infiltration presented as cytoplasmic vacuoles, while renal damage included tubular necrosis and glomerular alterations [23,62]. Finally, NAC demonstrated a protective role by attenuating hepatic and renal morphological damage in AFs-intoxicated birds [12,25], highlighting its potential as a chemoprotective agent. In the case of EQ, it also demonstrated a significant ability to reduce microscopic alterations in the liver and kidneys. However, few studies have evaluated its chemoprotective effect on these organs in animals intoxicated with aflatoxins, as EQ alone has been associated with adverse effects when used in various biological models. The European Food Safety Authority (EFSA) has reported that a dose of 50 mg of EQ/kg of feed is safe for laying hens [15]; however, the United States Food and Drug Administration considers a dose three times higher to be adequate [63]. In addition, the EFSA Panel on Additives and Products or Substances used in Animal Feed found that supplementing experimental diets with EQ is safe for the health and performance of laying hens, with a safety margin ten times higher than the recommended dose [15]. The main limitation to its use is the presence of the EQ precursor *p*-phenetidine, a possible mutagen, which could remain as an impurity in EQ and persist in animal tissues. Therefore, before recommending the extensive use of EQ, further studies would be necessary to assess the presence of this residue in EQ and in products derived from animals that have ingested it. In contrast, our study employed a dose of 330 mg of EQ/kg of feed (equivalent to EQ 33.3 mg/kg BW/d), and, like NAC, a reduction in hepatic and renal morphological alterations was observed in laying hens exposed to aflatoxins and treated with EQ.

## 4. Conclusions

The administration of aflatoxins at doses of 0.5, 1.0, and 1.5 mg/kg of feed over 72 weeks did not cause lethal toxicity in laying hens, indicating physiological resistance to prolonged exposure. The dietary supplementation with ethoxyquin or N-acetylcysteine was well tolerated and did not produce adverse effects in plasma or tissue biochemistry, nor did they cause macroscopic or histological alterations in the liver or kidneys of hens that did not consume aflatoxins. NAC and EQ showed chemoprotective activity by enhancing hepatic enzymatic and non-enzymatic detoxification pathways. This was evidenced by increased levels of reduced glutathione and elevated activity of GST and GGT enzymes. A significant reduction in biochemical markers of aflatoxicosis, such as ALT and AST, was observed, along with improved total protein concentrations in tissues and plasma. Moreover, the isolated use of these agents helped mitigate the hypertrophy of the liver and kidneys induced by chronic aflatoxin exposure. Histopathological analysis revealed less cellular degeneration in both organs, reinforcing the observed protective effect.

These findings demonstrate the effectiveness of NAC and EQ as mitigation strategies against aflatoxin toxicity and highlight their potential applicability in poultry production to enhance animal health and performance in environments where AFs contamination is prevalent. Although the absence of contaminants in the compounds used as chemoprotectants, AFs, and their persistence in animal tissues were not evaluated, this study provides evidence to support the development of novel feeding strategies, contributing to food security and animal welfare.

## 5. Materials and Methods

### 5.1. Experimental Birds and Housing

The Ethics Committee for the Use of Animals in Teaching and Research at the Autonomous University of Aguascalientes reviewed, authorized, and monitored this study to ensure compliance with the established protocol. The Committee justified the approval based on the high prevalence of AFs contamination detected in the region, particularly in poultry feed and eggs, which represents a considerable risk to both public and animal health. A total of 360-day-old Hy-Line W36 Leghorn hens were obtained from a local hatchery. Birds were selected based on similar body weight (66 ± 5.0 g) and absence of morphological defects. The animals were managed in accordance with standard production practices and fed a diet formulated according to the nutritional requirements for laying hens, without coccidiostats, antibiotics, or growth promoters [64]. In particular, the diet for laying hens was designed to satisfy the nutritional requirements of this phase [65], meeting or exceeding the following nutrient content: crude protein, linoleic acid, arginine, lysine, methionine, methionine + cysteine, tryptophan, threonine, calcium, total phosphorus, available phosphorus, sodium, chloride, potassium, and metabolizable energy (17.5, 1.5, 1.1, 0.88, 0.48, 0.82, 0.17, 0.68, 3.65, 0.65, 0.45, 0.18, 0.17, 0.50%, and 2940 kcal/kg). The diet for each treatment group was stored in airtight plastic containers (200 L) and kept at a controlled temperature, humidity, and ventilation.

Feed and water were provided *ad libitum* until week 85. During the rearing phase, birds were housed in battery cages with manual heating, and during the production phase, in two-level inverted pyramid cages. At 12 weeks of age, each hen was housed in an individual cage with a separate feeder, and the daily feed consumption was recorded for each bird. All animals were closely observed until 72 weeks of exposure. Body weight and egg production were recorded, and feed efficiency was calculated [5]. Temperature and relative humidity were maintained within the ranges recommended by the Federation of Animal Science Societies [64].

### 5.2. Experimental Design

Chronic intoxication was assessed using a factorial treatment design (Table 5) with one bird or experimental unit divided into 12 experimental treatment groups, each consisting of 30 individuals. Two dietary chemoprotection regimens were evaluated: NAC (80.8 mg/kg BW/d. N-Acetylcysteine, Pharmazell/AdyFarm, Mexico) and EQ (33.3 mg/kg BW/d; Dresquin-66-AQ, ethoxyquin at 66%, CFS-Dresen, Mexico), compared to a control group without chemoprotection, as well as four levels of aflatoxins (AFs: 0.0, 0.5, 1.0, and 1.5 mg/kg of feed). The evaluation of chemoprotective agents was performed by analyzing biochemical and pathological parameters at five slaughter time points (3 birds per group), corresponding to 0, 14, 20, 46, and 72 weeks of intoxication.

### 5.3. Production and Quantification of Total Aflatoxins

Corn immature grain samples of maize genotypes susceptible to colonization by toxigenic strains of *Aspergillus flavus* were used; these strains were cultured on potato dextrose agar and incubated in the dark at 27 °C for 10 days. Spores were harvested by washing with 0.1% Tween 20 solutions, generating a stock solution for inoculation. Corn grain samples were placed in flasks and inoculated under sterile, experimental conditions with 5 mL of inoculum (2.5 × 10^5^ spores/g of maize), moisture was adjusted to 15% using sterile distilled water, and spores were fixed with 1.0% paraffin oil. Paraffin oil was added to maize kernels to ensure homogeneous distribution and firm adhesion of spores in the corn. A control group of corn kernels was instilled with media but without spores. Incubation was carried out at 29 °C for 14 days with daily agitation to prevent adhesion [66]. To maintain AFs stability, contaminated dried corn batches were stored in hermetically sealed bags under freezing conditions (−20 °C) until use. The concentration of AFs in each batch of dry corn was estimated, and the corn was added to the basal diet according to the desired concentration for each treatment group. To ensure feed homogeneity, a trouser-type mixer was employed to progressively and uniformly blend the dietary components.

The basal and contaminated diets were analyzed by high-performance liquid chromatography for the AFs concentration, following AOAC Official Method 990.33 [67], achieving AFs concentrations of 0.0, 0.5, 1.0, and 1.5 mg/kg of feed. At weeks 0, 14, 20, 46, and 72, random samples from each batch were analyzed by HPLC, ensuring a maximum variation of ± 5.0% in the AFs concentration. Additionally, samples of contaminated and basal feed were analyzed for ochratoxin (OTA), fumonisins (FBs), zearalenone (ZEA), and deoxynivalenol (DON) using competitive ELISA kits (absorbance microplate reader, BioTek Instruments, Winooski, VT, USA; Ridascreen-Fast diagnostic kits for OTA, FBs, ZEA, and DON; R-Biopharm AG, Darmstadt, Germany). Standard curves were prepared from purified aflatoxins (B1, B2, G1, G2; Sigma Aldrich; OTA, FBs, ZEA, and DON, R-Biopharm AG, Darmstadt, Germany); all linear regression analyses yielded a coefficient of determination of R^2^ > 95.5% and the recoveries exceeded 87.5%. The detection limits for AFs, ZEA, OTA, DON, and FBs were 0.5, 29.0, 0.5, 20.0, and 30.0 µg/kg, respectively.

### 5.4. Collection of Biological Samples

Birds were weighed and anesthetized with sodium pentobarbital (25 mg/kg body weight, IV) until they exceeded a surgical plane. A midline ventral incision was made to expose the heart, and 180 mL of a washing solution (phosphate buffer, pH 7.4, heparin 500 U/L, procaine 1.0 g/L, 25 mL/min) was administered through the left cardiac ventricle for in situ perfusion. The liver and kidneys were then extracted and weighed to calculate their relative weight in relation to body weight. To quantify GSH in the liver and kidneys, 250 mg of tissue was homogenized immediately after collection in phosphate buffer (Na_2_HPO_4_: 14.19 g/L, EDTA: 1.86 g/L, pH: 8.0) with metaphosphoric acid (1.0 mL, 25%); the resulting homogenates were stored at −20 °C and analyzed within 24 h of processing. The determination of GST, GGT, and total protein levels in liver and kidney tissues, each sample (100 mg) was homogenized immediately after collection using a phosphate buffer (Na_2_HPO_4_: 14.19 g/L; pH: 6.5) and stored frozen (−20 °C) until analysis (24 h). For plasma protein quantification and enzymatic activity assessment of ALT and AST, 3 mL of intracardiac blood was collected using syringes containing 10% EDTA. Samples were centrifuged at 5000 rpm for 5 min to obtain plasma, which was stored at −20 °C until processing [68,69].

For histopathological evaluation, samples from the right hepatic lobe and right kidney were collected and preserved in a fixative solution (100 mL of 37% formaldehyde, 900 mL of distilled water, 4.0 g/L NaH_2_PO_4_, 6.5 g/L Na_2_HPO_4_, pH 7.4) until processing.

### 5.5. Biochemical and Histopathological Analysis

For GSH quantification, liver and kidney tissue homogenates were centrifuged (5000 rpm, 30 min), and 500 μL of the supernatant from each sample was mixed with 4.5 mL of phosphate buffer (pH 8.0). From this mixture, 100 μL were combined with 1.8 mL of phosphate buffer (pH 8.0) and 0.1 mL of o-phthalaldehyde (OPT 0.1% *w*/*v*) and incubated at room temperature for 15 min. Fluorescence was measured using a spectrofluorometer (Perkin Elmer LS50B, Norfolk, CA, USA) with excitation at 350 nm and emission at 420 nm [70]. The concentration of each sample was calculated using a standard curve based on a GSH reference solution.

To quantify specific enzymatic activity of hepatic and renal GST, a mixture was prepared using 2.3 mL of phosphate buffer (pH 6.5), 500 μL of GSH, 100 μL of tissue homogenate, and 100 μL of 1-chloro-2,4-dinitrobenzene (CDNB), the latter added under constant agitation. After resting for 15 min at room temperature, absorbance was read at 340 nm using a spectrophotometer (Varian DMS-80, Varian Ass., Inc., Sydney, Australia) [71]. For the quantification of total proteins and enzymatic activities of GGT, ALT, and AST, commercial spectrophotometric assay kits were used (Biosystems S. A., Leica Microsystem, Wetzlar, Germany).

Liver and kidney samples were subjected to routine histological processing, following the technique described by Prophet et al. [72]. The resulting tissue sections were stained with hematoxylin and eosin (H&E) using a semi-automatic linear stainer (Leica ST 4040, Leica Microsystem, Wetzlar, Germany). Preparations were examined under a compound microscope (Leica DM LS2, Leica Microsystem, Wetzlar, Germany). Hepatic steatosis was classified according to the extent of the lesion and the morphological characteristics of intracellular vacuolization [61]: grades 0, 1, 2, and 3, corresponding to <5%, 5–33%, 33–66%, and >66%, respectively.

### 5.6. Statistical Analysis

Data obtained from continuous variables (feed intake, hen-day egg production, egg weight, feed efficiency, GSH and total protein concentration, relative liver weight, specific enzyme activity of GST, GGT, ALT, and AST) were analyzed by ANOVA and Tukey’s HSD test to separate means between AFs groups (0.0, 0.5, 1.0, and 1.5 µg/kg) with or without NAC and EQ. A statistical comparison was made in weeks 0, 14, 20, 46, and 72 of the observation periods. Histological analysis was performed independently (blinded) by two pathologists and a third pathologist participated in discrepancies; for each specimen, only the result that matched between the two pathologists was recorded. The tissues were examined for a homogeneous distribution of lesions in the hepatic lobes or renal regions, so each specimen was reported as a single unit. Microscopic alterations were analyzed as categorical data (1 = lesion, 0 = no lesion detected) by estimating the proportion of lesions identified in 90 readings of 15 necropsies for each treatment; Bonferroni’s post hoc multiple comparison procedure was used [73]. A significant value of *p* < 0.05 was used for all tests. A posteriori statistical power was calculated according to the formula proposed by Brion et al. [74], considering the differences between the aflatoxin level groups, the sample size of each group, and an alpha error of 5%. Our analysis had a power of >75%.

## Figures and Tables

**Figure 1 toxins-17-00514-f001:**
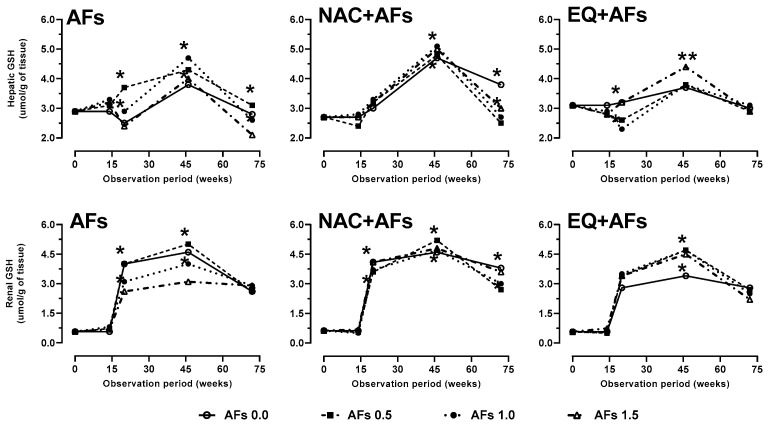
Effect of feed intake of N-acetylcysteine (NAC) or ethoxyquin (EQ) on reduced glutathione (GSH) concentration in the liver and kidneys of laying hens chronically exposed to aflatoxins (0.0–1.5 mg/kg). Asterisks (*, **) indicate significant differences (*p* < 0.05, *p* < 0.01) in at least one treatment in statistical comparison as a series of repeated measurements over time (weeks 0, 14, 20, 46, and 72) of the observation period.

**Figure 2 toxins-17-00514-f002:**
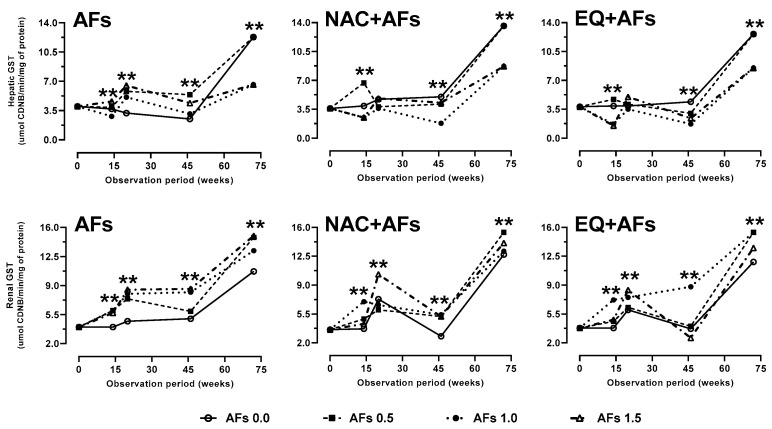
Effect of feed intake of N-acetylcysteine (NAC) or ethoxyquin (EQ) on the activity of glutathione S transferases (GST) in the liver and kidneys of laying hens chronically exposed to aflatoxins (0.0–1.5 mg/kg). Asterisks (**) indicate significant differences (*p* < 0.01) in at least one treatment in statistical comparison as a series of repeated measurements over time (weeks 0, 14, 20, 46, and 72) of the observation period.

**Figure 3 toxins-17-00514-f003:**
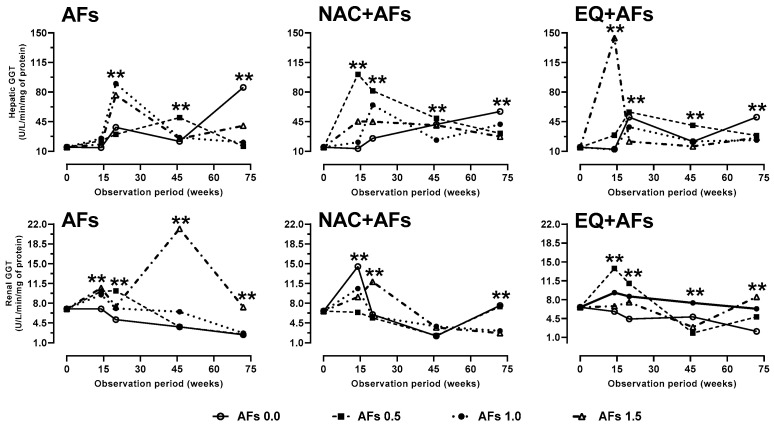
Effect of feed intake of N-acetylcysteine (NAC) or ethoxyquin (EQ) on the activity of gamma glutamyltransferase (GGT) in the liver and kidneys of laying hens chronically exposed to aflatoxins (0.0–1.5 mg/kg). Asterisks (**) indicate significant differences (*p* < 0.01) in at least one treatment in statistical comparison as a series of repeated measurements over time (weeks 0, 14, 20, 46, and 72) of the observation period.

**Figure 4 toxins-17-00514-f004:**
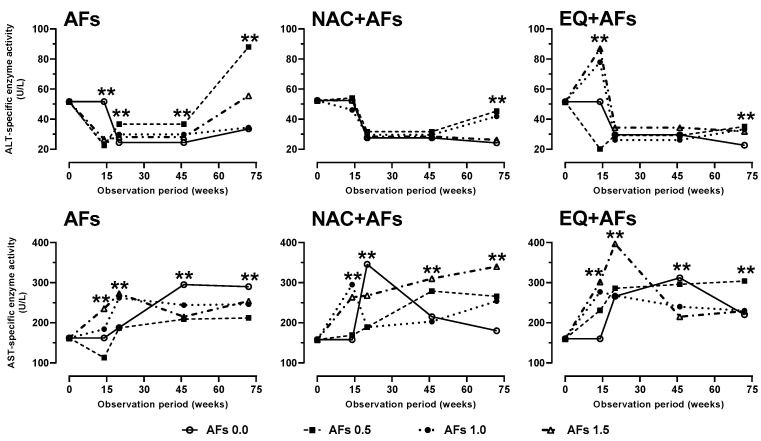
Effect of feed intake of N-acetylcysteine (NAC) or ethoxyquin (EQ) on the activity of alanine aminotransferase (ALT) and aspartate aminotransferase (AST) in plasma of laying hens chronically exposed to aflatoxins (0.0–1.5 mg/kg). Asterisks (**) indicate significant differences (*p* < 0.01) in at least one treatment in statistical comparison as a series of repeated measurements over time (weeks 0, 14, 20, 46, and 72) of the observation period.

**Figure 5 toxins-17-00514-f005:**
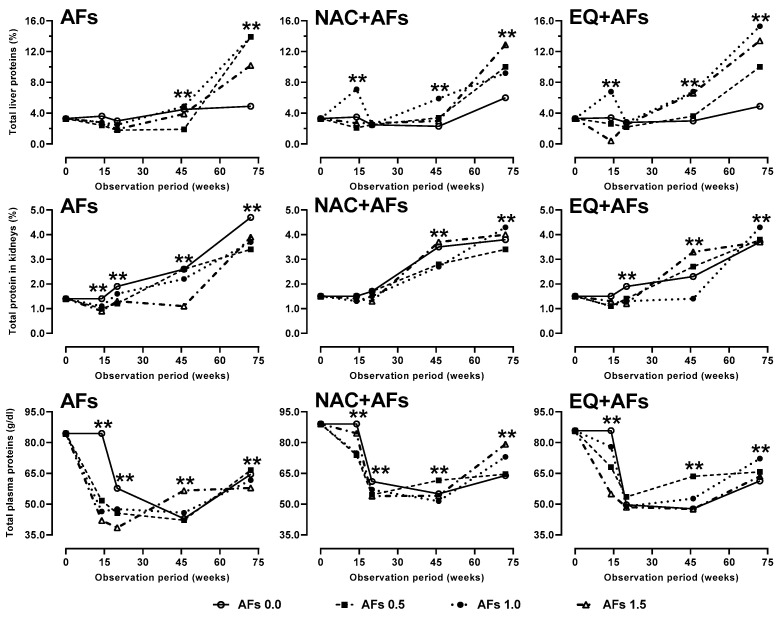
Effect of feed intake of N-acetylcysteine (NAC) or ethoxyquin (EQ) on total protein in the liver, kidneys, and plasma of laying hens chronically exposed to aflatoxins (0.0–1.5 mg/kg). Asterisks (**) indicate significant differences (*p* < 0.01) in at least one treatment in statistical comparison as a series of repeated measurements over time (weeks 0, 14, 20, 46, and 72) of the observation period.

**Figure 6 toxins-17-00514-f006:**
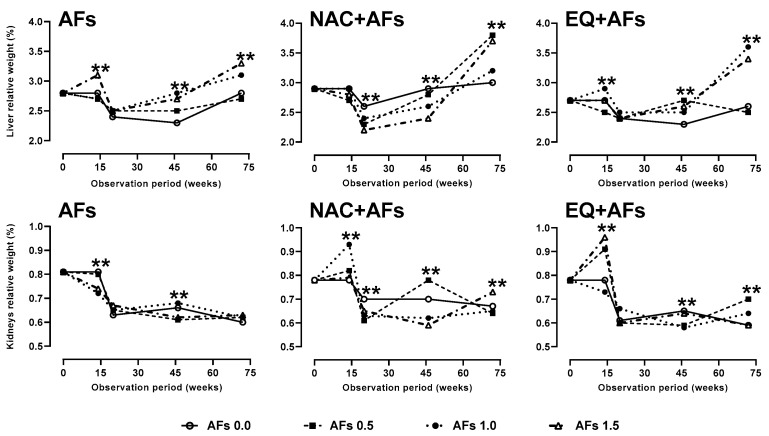
Effect of feed intake of N-acetylcysteine (NAC) or ethoxyquin (EQ) on the relative weight of the liver and kidneys of laying hens chronically exposed to aflatoxins (0.0–1.5 mg/kg). Asterisks (*, **) indicate significant differences (*p* < 0.05, 0.01) in at least one treatment in statistical comparison as a series of repeated measurements over time (weeks 0, 14, 20, 46, and 72) of the observation period.

**Figure 7 toxins-17-00514-f007:**
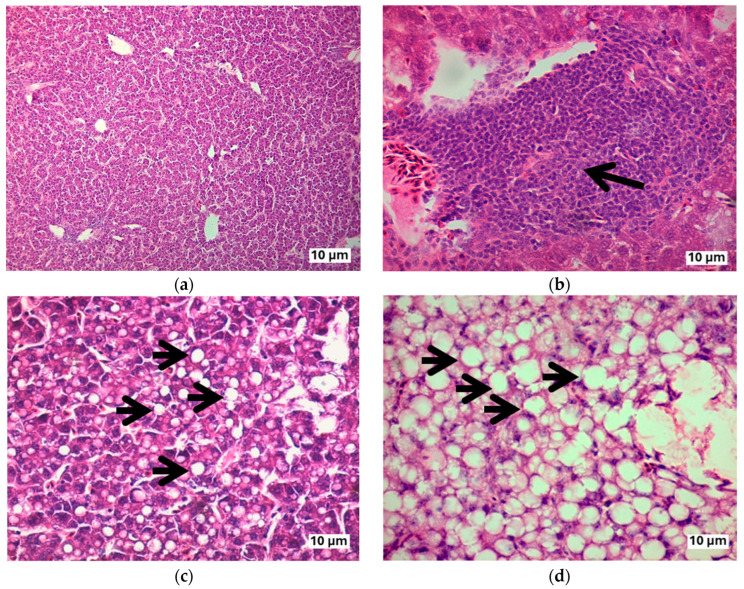
Toxic effect of aflatoxins on the liver of laying hens. (**a**) Control group: preserved hepatic architecture with no apparent morphological damage (40×); (**b**) Lymphocytic hepatitis: perivascular inflammatory infiltrate of lymphocytes (arrow; 400×); (**c**) Microvesicular steatosis: accumulation of small lipid droplets in the cytoplasm of hepatocytes (arrows), observed in hens after 14 weeks of intoxication (400×); (**d**) Macrovesicular steatosis: presence of large lipid droplets in the hepatocyte cytoplasm (arrows; 400×), observed in hens after 72 weeks of intoxication. H&E stain.

**Figure 8 toxins-17-00514-f008:**
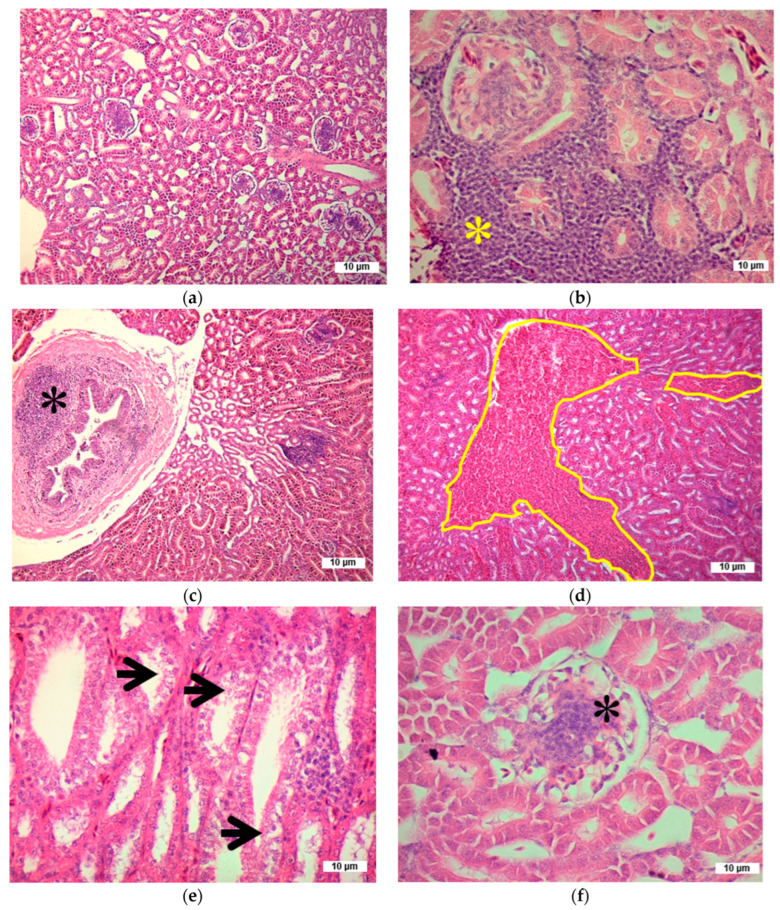
Toxic effect of aflatoxins on the kidneys of laying hens. (**a**) Control group: preserved renal structure with no apparent morphological alterations (40×); (**b**) Lymphocytic nephritis: focal lymphocyte inflammatory infiltrate (yellow asterisk; 400×); (**c**) Ureteritis: lymphocytic infiltrate in the lamina propria (asterisk; 40×); (**d**) Congestion: capillary dilation and erythrocyte accumulation in peritubular vessels (delimited area; 40×); (**e**) Tubular degeneration: loss of structural organization and epithelial desquamation into the tubular lumen (arrows; 400×); (**f**) Glomerular proliferation: increased number of mesangial cells (asterisk; 400×). H&E stain.

**Table 1 toxins-17-00514-t001:** Performance (mean ± SEM) of Hy-Line W36 Leghorn laying hens exposed to aflatoxins plus ethoxyquin or N-acetylcysteine in concentrated feed.

Performance	Time (wk)	Control *	Aflatoxins *	N-acetylcysteine *	Ethoxyquin *
Feed Intake (g/d)	14–20	81.2 ± 2.1 ^a^	80.0 ± 1.30 ^b^	80.0 ± 0.85 ^b^	80.8 ± 1.87 ^ab^
	21–46	112 ± 1.61 ^a^	98.4 ± 1.35 ^b^	112 ± 2.1 ^a^	111 ± 2.2 ^a^
	47–72	104 ± 3.7 ^a^	96.2 ± 1.95 ^b^	104 ± 4.3 ^a^	103 ± 1.72 ^a^
Hen-day egg production (%) **	14–20	78.6 ± 3.7 ^a^	65.3 ± 2.8 ^b^	78.6 ± 1.56 ^a^	65.7 ± 1.70 ^b^
	21–46	83.6 ± 1.62 ^c^	88.6 ± 1.28 ^b^	88.6 ± 0.59 ^b^	89.8 ± 0.45 ^a^
	47–72	81.3 ± 1.08 ^b^	79.4 ± 1.07 ^c^	79.9 ± 1.21 ^c^	84.8 ± 0.53 ^a^
Egg weight (g/egg)	14–20	58.1 ± 0.75 ^c^	59.6 ± 1.08 ^a^	59.6 ± 0.59 ^a^	58.9 ± 0.83 ^b^
	21–46	64.5 ± 1.96 ^c^	65.9 ± 1.08 ^b^	65.9 ± 1.08 ^b^	68.1 ± 0.28 ^a^
	47–72	67.0 ± 0.26 ^b^	67.3 ± 0.82 ^b^	67.0 ± 0.82 ^b^	69.3 ± 1.62 ^a^
Feed efficiency (Laying/feed; kg/ton)	14–20	543 ± 26.9 ^b^	473 ± 28.1 ^c^	587 ± 26.9 ^a^	480 ± 26.9 ^c^
	21–46	485 ± 24.3 ^d^	591 ± 17.4 ^a^	522 ± 13.2 ^c^	551 ± 12.3 ^b^
	47–72	527 ± 25.2 ^b^	559 ± 12.8 ^a^	518 ± 24.8 ^b^	570 ± 15.9 ^a^

* Dosage: Control: Basal diet. Aflatoxins: 1.5 + mg/kg of feed. N-acetylcysteine and Ethoxyquin: 80.8 and 33.3 mg/kg BW/d, respectively. ** Average of the total number of eggs laid per hen, divided by the number of days in the period, and multiplied by 100. ^a–d^ Means with different literal are significantly different in Tukey’s HSD test; *p*-value < 0.05.

**Table 2 toxins-17-00514-t002:** Effect of feed intake of N-acetylcysteine, Ethoxyquin, and their interaction on *p*-value in general linear model analysis during chronic exposure (72 weeks) to aflatoxins (AFs) in laying hens.

Week	Effect	Liver				Kidneys				Plasma		
		Prot	GSH	GST	GGT	Prot	GSH	GST	GGT	Prot	ALT	AST
**N-acetylcysteine (NAC)**												
14	Model	0.067	0.163	0.001	0.036	0.017	0.000	0.001	0.016	0.000	0.000	0.000
	NAC	0.373	0.990	0.472	0.314	0.908	0.124	0.392	0.102	0.109	0.085	0.385
	AFs	0.266	0.701	0.007	0.013	0.002	0.001	0.000	0.006	0.000	0.013	0.000
	NACxAFs	0.038	0.178	0.010	0.114	0.829	0.000	0.595	0.069	0.006	0.037	0.314
20	Model	0.609	0.001	0.409	0.285	0.001	0.000	0.000	0.000	0.000	0.966	0.032
	NAC	0.482	0.014	0.188	0.207	0.540	0.505	0.016	0.544	0.000	0.990	0.006
	AFB	0.264	0.119	0.817	0.325	0.053	0.006	0.000	0.000	0.080	0.741	0.609
	NACxAFs	0.505	0.898	0.100	0.729	0.031	0.004	0.068	0.002	0.003	0.813	0.006
46	Model	0.011	0.005	0.000	0.045	0.206	0.019	0.138	0.000	0.000	0.966	0.032
	NAC	0.345	0.274	0.219	0.496	0.080	0.010	0.365	0.000	0.030	0.990	0.006
	AFs	0.002	0.002	0.000	0.007	0.519	0.489	0.040	0.546	0.007	0.741	0.609
	NACxAFs	0.708	0.840	0.490	0.455	0.046	0.286	0.256	0.000	0.000	0.813	0.006
72	Model	0.875	0.374	0.000	0.014	0.004	0.002	0.196	0.001	0.050	0.202	0.000
	NAC	0.538	0.696	0.198	0.389	0.154	0.147	0.622	0.014	0.706	0.234	0.002
	AFs	0.832	0.371	0.000	0.003	0.021	0.030	0.068	0.092	0.424	0.840	0.001
	NACxAFs	0.805	0.631	0.923	0.211	0.009	0.556	0.326	0.000	0.073	0.893	0.000
**Ethoxyquin (EQ)**												
14	Model	0.719	0.020	0.000	0.000	0.006	0.006	0.050	0.498	0.000	0.000	0.000
	EQ	0.537	0.559	0.120	0.204	0.815	0.059	0.573	0.453	0.188	0.495	0.113
	AFs	0.343	0.643	0.010	0.001	0.008	0.011	0.008	0.364	0.000	0.143	0.000
	EQxAFs	0.674	0.020	0.001	0.002	0.079	0.753	0.871	0.239	0.135	0.001	0.190
20	Model	0.057	0.789	0.000	0.000	0.001	0.000	0.000	0.557	0.000	0.763	0.000
	EQ	0.853	0.538	0.887	0.021	0.571	0.000	0.392	0.997	0.048	0.738	0.008
	AFs	0.048	0.451	0.000	0.202	0.000	0.005	0.000	0.196	0.000	0.336	0.000
	EQxAFs	0.176	0.518	0.029	0.000	0.294	0.000	0.203	0.727	0.000	0.666	0.710
46	Model	0.000	0.055	0.008	0.358	0.051	0.003	0.011	0.000	0.012	0.966	0.045
	EQ	0.363	0.094	0.327	0.749	0.183	0.006	0.753	0.204	0.003	0.990	0.060
	AFs	0.001	0.035	0.274	0.232	0.188	0.294	0.052	0.001	0.150	0.741	0.977
	EQxAFs	0.005	0.388	0.007	0.595	0.021	0.000	0.117	0.001	0.022	0.813	0.007
72	Model	0.003	0.000	0.000	0.032	0.325	0.000	0.155	0.000	0.026	0.060	0.775
	EQ	0.327	0.618	0.876	0.225	0.159	0.065	0.637	0.339	0.327	0.074	0.699
	AFs	0.000	0.004	0.000	0.007	0.342	0.185	0.028	0.000	0.423	0.581	0.345
	EQxAFs	0.195	0.007	0.185	0.456	0.117	0.000	0.716	0.203	0.015	0.848	0.823

Prot = Proteins; GSH = Reduced glutathione; GST = Glutathione S-transferases; GGT = Gamma glutamyltransferase; ALT = Alanine aminotransferase: AST = aspartate aminotransferase.

**Table 3 toxins-17-00514-t003:** Effect of N-acetylcysteine (NAC) and Ethoxyquin (EQ) on the frequency of liver injury caused by chronic ingestion of aflatoxins (AFs) in laying hens.

Treatment (*)				Microscopic Alteration (**)						Proportion of Lesions (***)
AFs	EQ	NAC		Lymp	HydrD	Steat	Hem	Other	Total	
0.0	0.0	0.0		0	0	0	0	0	0	0.0 (0.0–7.3 ^d^)
0.0	33.3	0.0		0	0	0	0	0	0	0.0 (0.0–7.3 ^d^)
0.0	0.0	80.8		0	0	0	0	0	0	0.0 (0.0–7.3 ^d^)
0.5	0.0	0.0		0	0	0	0	0	0	0.0 (0.0–7.3 ^d^)
0.5	33.3	0.0		0	0	0	0	0	0	0.0 (0.0–7.3 ^d^)
0.5	0.0	80.8		0	0	0	0	0	0	0.0 (0.0–7.3 ^d^)
1.0	0.0	0.0		15	0	0	0	0	15	20.0 (12.7–27.3 ^bc^)
1.0	33.3	0.0		4	0	0	0	0	4	5.3 (0.0–12.7 ^cd^)
1.0	0.0	80.8		9	0	0	0	0	9	12.0 (4.7–19.3 ^cd^)
1.5	0.0	0.0		15	15	15	7	7	59	78.7 (71.3–86.0 ^a^)
1.5	33.3	0.0		15	4	2	0	0	21	28.0 (20.7–35.3 ^b^)
1.5	0.0	80.8		15	0	0	0	0	15	20.0 (12.7–27.3 ^bc^)
Total				73	19	17	7	7	123	

* AFs: µg/kg feed. EQ and NAC: µg/kg BW. ** Lymp = Lymphocytosis; HydrD = Hydropic Degeneration; Steat = Steatosis; Hem = Hemorrhage; Other = necrotic foci, proliferation of bile ducts. *** Proportion of lesions identified in 75 readings from 15 necropsies for each treatment. ^a–d^ Bonferroni post hoc multiple comparison procedure and 95% confidence intervals; *p*-value < 0.05.

**Table 4 toxins-17-00514-t004:** Effect of N-acetylcysteine (NAC) and Ethoxyquin (EQ) on the frequency of kidney injury caused by chronic ingestion of aflatoxins (AFs) in laying hens.

Treatment (*)				Microscopic Alteration (**)							Proportion of Lesions (***)
AFs	EQ	NAC		Con	Linf	Ure	Thro	Hem	Other	Total	
0.0	0.0	0.0		0	0	0	0	0	0	0	0.0 (0.0–7.4 ^e^)
0.0	33.3	0.0		0	0	0	0	0	0	0	0.0 (0.0–7.4 ^e^)
0.0	0.0	80.8		0	0	0	0	0	0	0	0.0 (0.0–7.4 ^e^)
0.5	0.0	0.0		0	4	0	0	0	0	4	4.4 (0.0–11.8 ^de^)
0.5	33.3	0.0		0	0	0	0	0	0	0	0.0 (0.0–7.4 ^e^)
0.5	0.0	80.8		0	0	0	0	0	0	0	0.0 (0.0–7.4 ^e^)
1.0	0.0	0.0		0	15	0	0	0	6	21	16.7 (9.3–24.1 ^d^)
1.0	33.3	0.0		0	15	0	0	0	0	15	16.7 (9.3–24.1 ^d^)
1.0	0.0	80.8		0	15	0	0	0	0	15	16.7 (9.3–24.1 ^d^)
1.5	0.0	0.0		15	15	15	15	12	8	80	88.9 (81.5–96.3 ^a^)
1.5	33.3	0.0		9	15	15	7	0	0	46	51.1 (43.7–58.5 ^b^)
1.5	0.0	80.8		4	15	11	0	0	0	30	33.3 (25.9–40.7 ^c^)
Total				28	94	41	22	12	14	211	

* AFs: µg/kg feed. EQ and NAC: µg/kg BW. ** Con = Kidney congestion; Lymp = Lymphocytic nephritis; Ure = Ureteritis; Thro = Renal thrombosis; Hem = Hemorrhages; Others = Glomerular proliferation, glomerulonephritis, tubular degeneration, detachment of the brush border in renal tubules, and necrotic foci. *** Proportion of lesions identified in 90 readings from 15 necropsies for each treatment. ^a–e^ Bonferroni post hoc multiple comparison procedure and 95% confidence intervals; *p*-value < 0.05.

**Table 5 toxins-17-00514-t005:** Experimental treatment groups for chemoprotection against aflatoxin exposure in laying hens *.

Dietary Regimen **	Treatment Group	AFs	NAC	EQ
		mg/kg of Feed	mg/kg BW/d	mg/kg BW/d
AFs	T_1_: AFs 0.0	0.00	0.00	0.00
	T_2_: AFs 0.5	0.50	0.00	0.00
	T_3_: AFs 1.0	1.00	0.00	0.00
	T_4_: AFs 1.5	1.50	0.00	0.00
NAC	T_5_: NAC + 0.5 AFs	0.00	80.8	0.00
	T_6_: NAC + 0.5 AFs	0.50	80.8	0.00
	T_7_: NAC + 1.0 AFs	1.00	80.8	0.00
	T_8_: NAC + 1.5 AFs	1.50	80.8	0.00
EQ	T_9_: EQ + 0.5 AFs	0.00	0.00	33.3
	T_10_: EQ + 0.5 AFs	0.50	0.00	33.3
	T_11_: EQ + 1.0 AFs	1.00	0.00	33.3
	T_12_: EQ + 1.5 AFs	1.50	0.00	33.3

* 360-Hy-Line W36 Leghorn laying hens and 30 hens per treatment group, ** AFs = Aflatoxins, NAC = N-acetylcysteine, EQ = Ethoxyquin. T_1_–T_12_ Experimental treatment groups.

## Data Availability

The original contributions presented in this study are included in the article. Further inquiries can be directed to the corresponding author(s).

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
