# Peer review of "Protective Effect of Ethoxyquin and N-acetylcysteine on Biochemical and Pathological Changes Induced by Chronic Exposure to Aflatoxins in Laying Hens"

_toxins, 2025, doi:10.3390/toxins17100514_

Round 1

Reviewer 1 Report

Comments and Suggestions for Authors

The manuscript explores the chemoprotective effects of N-acetylcysteine (NAC) and ethoxyquin (EQ) against chronic aflatoxicosis in laying hens. The study is ambitious and of potential importance to poultry science and food safety. However, several methodological and interpretational issues significantly weaken the conclusions. I recommend a major revision before further consideration.

Major Comments

  1. Feed preparation and mycotoxin control

AFs quantification by HPLC (AOAC 990.33) is acceptable, but authors should provide validation parameters (LOD/LOQ, recoveries, linearity).

Exclusion of other mycotoxins using only ELISA is insufficient. Multi-mycotoxin LC–MS/MS analysis (DON, ZEA, OTA, FBs, cyclopiazonic acid, etc.) is necessary, at least to confirm negative results in each batch and at each time point (0–72 weeks).

Please include data on feed homogeneity, AFs stability during storage, and actual feed intake.

Paraffin addition (1%) for spore fixation could influence lipid metabolism. Please clarify whether paraffin was added equally to all groups, including controls.Dosage and justification of chemoprotectants

The EQ dose (500 mg/kg) is 10× above the EFSA safe level for laying hens (50 mg/kg). This requires strong toxicological justification and discussion of potential residues in eggs/tissues.

The NAC dose (800 mg/kg) should be expressed in terms of daily intake (mg/kg BW/day) and compared with literature. Biomarkers of compliance (NAC/cysteine in plasma) would strengthen conclusions.

No data on residues of AF metabolites or EQ/NAC in eggs are provided. This omission limits the food safety relevance of the study.

  1. Histopathology

Figure 7b (“Lymphocytic hepatitis…”) is difficult to interpret at the current magnification. Please provide higher magnifications (×200/×400) with clear arrows and scale bars.

Differentiating steatosis from glycogen vacuolation or hydropic change is problematic in H&E. Oil Red O or PAS with or without diastase staining (or at least a limitation statement) should be added.

Lesion scoring is unclear. Please define the scale (0–3, 0–4) and use appropriate statistical methods such as ordinal logistic regression or GLMM. Percentages should not be treated as continuous variables without clear denominators.

  1. Statistical analysis and experimental unit

Sampling was from different birds at each time point, so repeated measures ANOVA is inappropriate. Time should be treated as a between-subject factor, or mixed models with pen as the random effect should be employed.

Clarify the experimental unit for production data (pen versus individual hen).

Provide a post hoc power analysis—sample size (n=3 per group/time) is likely underpowered.

Consider correction for multiple comparisons. External relevance and food safety

Chronic exposure to 1.0–1.5 mg/kg AFs over 72 weeks is significantly higher than typical field conditions. Please discuss its real-world relevance. The lack of data on residues in eggs (AFB1/AFM1, EQ metabolites) limits the applicability of the findings. Acknowledging this as a major limitation would be important.

Minor Comments

ALT/AST are plasma enzymes, not tissue markers. Please correct the Figure 4 legend and text.

Table 2: correct “AFB” to “AFs”.

Results on total protein are inconsistent (initially described as decreased, later increased compared to control at 72 weeks). Please clarify.

Please ensure all histological figures display clear scale bars.

Improve English style in some sections (“good tolerance without adverse effects” for EQ is misleading at 10× the safe dose).

Author Response

The authors sincerely thank you for your suggestions and comments, which significantly improved the manuscript. We also appreciate your patience in reading and trying to understand the text in detail, as well as your ability to offer helpful comments. Below, we inform you how we have addressed your observations and comments.

Major Comments

  1. Feed preparation and mycotoxin control

1.1. AFs quantification by HPLC (AOAC 990.33) is acceptable, but authors should provide validation parameters (LOD/LOQ, recoveries, linearity).

Parameters for validating mycotoxin quantification were included in the materials and methods section of the manuscript (Lines: 533-538). Standard curves were prepared from purified aflatoxins, and all linear regression analyses yielded a coefficient of determination of R2 > 95.5%; the AFs recovery exceeded 90.5%, and the detection and quantification limits for AFs were 0.5 µg/kg and 1.5 µg/kg, respectively.

1.2. Exclusion of other mycotoxins using only ELISA is insufficient. Multi-mycotoxin LC–MS/MS analysis (DON, ZEA, OTA, FBs, cyclopiazonic acid, etc.) is necessary, at least to confirm negative results in each batch and at each time point (0–72 weeks).

An explanation of the methods used to quantify mycotoxins in this study was included in the discussion section of this manuscript (Lines 297-313). Although LC-MS/MS is currently widely used as a confirmatory reference method for the simultaneous determination of different mycotoxins, in this study, the quantification of the concentration of non-aflatoxin mycotoxins was performed using ELISA, following extraction, cleanup on immunoaffinity columns, and comparison against the results of purified mycotoxin standards. This ELISA method for the determination of mycotoxins is acceptable to several authors, who agree that there is a very high correlation between the concentrations of mycotoxins in feed and animal products, estimated comparatively between ELISA and LC-MS/MS. Therefore, it can be assumed that both techniques are satisfactory in terms of meeting feed safety control requirements.

As in other studies on natural mycotoxin contamination in hen feed, low concentrations of ZEN, OTA, DON, and FB were found. The presence of FB, DON, and OTA was below the minimum detection limits, while ZEN concentration was only detected occasionally during weeks 14 and 46 of the study (63.1 ± 24.4 μg/kg). The concentration of ZEN in feed that could cause a decrease in bird performance is >50 mg/kg; however, in this study, the observed ZEN concentration was several times lower than that concentration, suggesting that the health changes detected in this study were not caused by ZEN or other mycotoxins.

1.3. Please include data on feed homogeneity, AFs stability during storage, and actual feed intake.

An explanation of the methods used to quantify mycotoxins in this study was included in the Materials and Methods section of this manuscript (Lines 519-524).  In this description it is pointed out that to maintain AFs stability, contaminated dried corn batches were stored in hermetically sealed bags under freezing conditions (-20°C) until use. The concentration of AFs in each batch of dry corn was estimated, and the corn was added to the basal diet according to the desired concentration for each treatment group. To ensure feed homogeneity, a trouser-type mixer was employed to progressively and uniformly blend the dietary components.

1.4. Paraffin addition (1%) for spore fixation could influence lipid metabolism. Please clarify whether paraffin was added equally to all groups, including controls.

(Lines 515-518) Paraffin oil was added only to maize kernels for AFs production (5 mL of inoculum with 1.0% paraffin oil, sterile distilled water, and spores) to ensure homogeneous distribution and firm adhesion of spores in the corn. A control group of corn kernels was instilled with media but without spores. The contaminated corn was added to the basal diet according to the desired concentration for each treatment group.

  1. Dosage and justification of chemoprotectants

2.1. The EQ dose (500 mg/kg) is 10× above the EFSA safe level for laying hens (50 mg/kg). This requires strong toxicological justification and discussion of potential residues in eggs/tissues.

In the discussion section of the manuscript (Lines 434-442), it has been established that the EFSA has reported a dose of 50 mg EQ/kg feed is safe for laying hens; however, the US FDA considers a dose three times higher to be adequate. In addition, the EFSA Panel on Additives and Products or Substances used in Animal Feed has determined that in laying hens, there is a safety margin ten times higher than the recommended dose. The main limitation to its use is the presence of the EQ precursor p-phenetidine, a possible mutagen, which could remain as an impurity in EQ and persist in animal tissues. Therefore, before recommending the extensive use of EQ, this study acknowledges that further studies would be necessary to assess the presence of this residue in EQ and in products derived from animals that have ingested it.

To make the expression of the EQ dose used comparable to the suggested adjustment for NAC, the dose was expressed in mg per kilogram of body weight, and the respective adjustments were made throughout the document. A modification of the EQ dose was also made, because the product used had an inert base that diluted the compound to 66%.

2.2. The NAC dose (800 mg/kg) should be expressed in terms of daily intake (mg/kg BW/day) and compared with literature. Biomarkers of compliance (NAC/cysteine in plasma) would strengthen conclusions.

The NAC dose was expressed in mg per kilogram of body weight by day, and the respective adjustments were made throughout the document. In the discussion section of the manuscript (Lines 322-325), it has been established that the NAC use in birds has been developed previously, and the protective properties have been shown; This confirms what is stated in the introduction section of the manuscript (Lines 61- 66).

2.3. No data on residues of AF metabolites or EQ/NAC in eggs are provided. This omission limits the food safety relevance of the study.

In the conclusion section of the manuscript (Lines 462-465), we acknowledge this lack of information because we agree with the observation. Another study is needed in short to clarify the presence of residues in eggs and chicken meat, both of AFs and EQ precursors. Although these metabolites were not evaluated, this study provides evidence to support the development of novel feeding strategies, contributing to food security and animal welfare.

  1. Histopathology

3.1. Figure 7b (“Lymphocytic hepatitis…”) is difficult to interpret at the current magnification. Please provide higher magnifications (×200/×400) with clear arrows and scale bars.

We have attended to your recommendation and updated Figure 7b in the manuscript, incorporating a higher magnification (400X), indicative arrows, and clear scale bars for easier reading.

3.2. Differentiating steatosis from glycogen vacuolation or hydropic change is problematic in H&E. Oil Red O or PAS with or without diastase staining (or at least a limitation statement) should be added.

In the discussion section of the manuscript (Lines 420-425), the cellular vacuolation is analyzed when it was detected using hematoxylin and eosin staining, although the characterization was performed with sufficient precision to identify it as compatible with hepatic steatosis. Although no distinction was made between lipid accumulation and glycogen vacuolization or hydropic changes. It is important to note that H&E staining remains a widely accepted method for identifying both microvesicular and macrovesicular hepatic steatosis.

3.3. Lesion scoring is unclear. Please define the scale (0–3, 0–4) and use appropriate statistical methods such as ordinal logistic regression or GLMM. Percentages should not be treated as continuous variables without clear denominators.

An explanation of the methods used to evaluate hepatic steatosis in this study was included in the materials and methods section of this manuscript (Lines 578-580, 595-598). Hepatic steatosis was classified according to the extent of the lesion and the morphological characteristics of intracellular vacuolization [61]: grade 0, 1, 2, and 3 with <5, 5-33, 33-66, and > 66%. However, the best presentation of the lesion's behavior was when we used the simple frequency of presence (or absence) of hepatic steatosis. The various lesions detected were summed for each treatment, and the frequency of lesions per treatment was expressed as a percentage of the samples analyzed and statistically analyzed as a continuous variable

  1. Statistical analysis and experimental unit

4.1. Sampling was from different birds at each time point, so repeated measures ANOVA is inappropriate. Time should be treated as a between-subject factor, or mixed models with pen as the random effect should be employed.

An explanation of the methods used to evaluate the variables in this study was included in the Statistical Analysis subsection of this manuscript (Lines 503-506, 282-286). Data obtained from continuous variables were analyzed in weeks 0, 14, 20, 46, and 72 of the observation periods by ANOVA and Tukey's HSD test to separate means among the twelve treatment groups.

4.2. Clarify the experimental unit for production data (pen versus individual hen).

Only one hen was housed per cage, but in this case the bird was defined as the experimental unit (Lines 493-496).

4.3. Provide a post hoc power analysis—sample size (n=3 per group/time) is likely underpowered.

In the statistical analysis section of the manuscript, the procedure for calculating the statistical power of the tests for continuous variables and frequency of lesions, the value achieved, and the bibliographic reference corresponding to this procedure have been added.

  1. External relevance and food safety

5.1. Chronic exposure to 1.0–1.5 mg/kg AFs over 72 weeks is significantly higher than typical field conditions. Please discuss its real-world relevance.

In the discussion section of the manuscript (Lines 314-320), it has been established that in several countries, data on poultry feed contamination is limited; however, studies indicate that AFs contamination in feed is common, making chronic exposure in vulnerable production systems a plausible concern. Previous studies have shown that normal homeostasis in laying hens continues to function regularly, even at doses as high as 1.5 mg of AFs per kg of body weight. Therefore, the high levels of aflatoxins used in this study were selected to challenge the efficacy of chemoprotectors, even in situations of high exposure in regions with limited quality control measures.

5.2. The lack of data on residues in eggs (AFB1/AFM1, EQ metabolites) limits the applicability of the findings. Acknowledging this as a major limitation would be important.

In the conclusion section of the manuscript (Lines 462-465), we acknowledge this lack of information because we agree with the observation. Another study is needed in short to clarify the presence of residues in eggs and chicken meat, both of AFs and EQ precursors. Although these metabolites were not evaluated, this study provides evidence to support the development of novel feeding strategies, contributing to food security and animal welfare.

  1. Minor Comments

6.1. ALT/AST are plasma enzymes, not tissue markers. Please correct the Figure 4 legend and text.

In Figure 4 title was modified to include the word plasma. In addition, in the Discussion section (Lines 389-391), it is described that ALT and AST are enzymes measured in plasma, but their biochemical function consists of introducing the amino acids alanine or aspartate from plasma through the cell wall into the cytosol. Therefore, when the membrane is damaged, AST/ALT increases their specific enzymatic activity in plasma, suggesting cell damage.

6.2. Table 2: correct “AFB” to “AFs”.

The correction requested in Table 2, changing “AFB” to “AFs- has been implemented in the revised version of the manuscript.

6.3. Results on total protein are inconsistent (initially described as decreased, later increased compared to control at 72 weeks). Please clarify.

Hens exposed to AFs showed a significant decrease (P<0.01) in plasma proteins until week 46, followed by stabilization. Protein concentration in hen tissues increased significantly at 72 weeks, with levels surpassing those of the treatment group with 0.0 mg AFs/kg. This trend was also observed in hens treated with NAC and EQ alongside AFs. In the kidney, protein concentration in AFs-intoxicated hens increased over time, although it remained significantly reduced (P<0.01) compared to the 0.0 mg AFs/kg). In contrast, in the NAC and EQ groups, renal protein levels remained stable in the second half of the study (Lines 189-198).

6.4. Please ensure all histological figures display clear scale bars.

All histological figures have been reviewed and updated to include clear scale bars in the revised version of the manuscript.

6.5. Improve English style in some sections (“good tolerance without adverse effects” for EQ is misleading at 10× the safe dose)

The entire manuscript was reviewed again, and several sections were improved, especially the conclusions. All changes are highlighted in yellow.

Reviewer 2 Report

Comments and Suggestions for Authors

The manuscript toxins-3845971, with the title “Protective effect of ethoxyquin and N-acetylcysteine on bio-chemical and pathological changes induced by chronic exposure to aflatoxins in laying hens”, investigated the effects of dietary supplementation with either N-acetylcysteine (NAC) or ethoxyquin (EQ) in laying hens exposed to aflatoxins (AFs) contaminated diet for period of up to 72 weeks. The study is very complex, supported by an excellent statistical pre-analysis that accounts for both the concentration of AFs and the exposure time of laying hens to the contaminated diet, as well as the effects of NAC and EQ supplementation. Furthermore, the study presents new elements regarding alterations in plasma, cellular biochemistry and histology, and the chemoprotective effects of NAC and EQ. These findings are important for understanding the chronic effects of AFs exposure in laying hens. It is appreciable that the authors used AFs concentrations detected in the area where the study was conducted, which represents a significant risk to both public and animal health. This real-world relevance strengthens the impact of the study and shows how such investigations should be conducted!

Below are my comments and suggestions for improving the manuscript:

-Section 2: Table 5 clearly shows that NAC and EQ were administered concomitantly with AFs. However, this is not evident from the graphs in the results section. It may be more informative to label the graphs as “AFs + NAC or EQ” rather than simply NAC or EQ.

-Section 3: How were the NAC and EQ concentrations selected for dietary supplementation? The rationale for choosing these concentrations should be addressed in the discussion section.

-Section 3: How do the authors explain the differences observed in hepatic GSH, hepatic GGT, ALT, and relative liver and kidney weights when NAC or EQ is administered alone compared with the control diet without AFs (0.0 mg/kg)? These differences should be addressed in the discussions section. In addition, is there statistical significance for the comparisons between NAC alone vs. control diet and EQ alone vs. control diet?

-Section 5.1: The number of laying hens per cage should be specified.

-Section 5.2: What were the concentrations of ochratoxin (OTA), fumonisins (FBs), zearalenone (ZEA), and deoxynivalenol (DON) determined using competitive ELISA kits? Details regarding the characterization of the diet should also be provided, particularly the physicochemical parameters. These could be briefly described in the methodology section or presented in a separate subsection of the results. These data can help other researchers in interpreting these results or in reproducing the experiment in future results.

-Section 5.3: Were the kidney and liver samples for GSH, GST, GGT, and total protein analysis homogenized immediately after collection? Were the tissue samples or homogenates stored at -20 °C or -80 °C prior to analysis? The description of the sample processing for these parameters is unclear and should be specified.

-Section 5.5: It may be clearer if the section about experimental design is placed at the beginning of the methodology, so that the presentation follows the actual sequence of the study.

-Section 5.6: Whether data from technical repetitions were averaged?

-Section 5.6: What was the power of used statistical tests?

Minor suggestions:

-Avoid using the long dash (—) in phrases such as “A significant reduction in biochemical markers of aflatoxicosis—such as ALT and AST—was observed”. This style is often characteristic of AI generated text.

-Section 2: Consider shortening the explanations in the results section, since similar points are already covered in the discussions section.

Author Response

The authors sincerely thank you for your suggestions and comments, which significantly improved the manuscript. We also appreciate your patience in reading and trying to understand the text in detail, as well as your ability to offer helpful comments. Below, we inform you how we have addressed your observations and comments.

  1. -Section 2: Table 5 clearly shows that NAC and EQ were administered concomitantly with AFs. However, this is not evident from the graphs in the results section. It may be more informative to label the graphs as “AFs + NAC or EQ” rather than simply NAC or EQ.

The graphs in the results section have been updated in the revised version of the manuscript to more clearly reflect the concomitant administration of AF with NAC or EQ, using more informative labels such as “AFs+NAC” and “AFs+EQ”.

  1. -Section 3: How were the NAC and EQ concentrations selected for dietary supplementation? The rationale for choosing these concentrations should be addressed in the discussion section.

In the discussion section of the manuscript (Lines 434-442), it has been established that the EFSA has reported a dose of 50 mg EQ/kg feed is safe for laying hens; however, the US FDA considers a dose three times higher to be adequate. In addition, the EFSA Panel on Additives and Products or Substances used in Animal Feed has determined that in laying hens, there is a safety margin ten times higher than the recommended dose. The main limitation to its use is the presence of the EQ precursor p-phenetidine, a possible mutagen, which could remain as an impurity in EQ and persist in animal tissues. Therefore, before recommending the extensive use of EQ, this study acknowledges that further studies would be necessary to assess the presence of this residue in EQ and in products derived from animals that have ingested it.

To make the expression of the EQ dose used comparable to the suggested adjustment for NAC, the dose was expressed in mg per kilogram of body weight, and the respective adjustments were made throughout the document. A modification of the EQ dose was also made, because the product used had an inert base that diluted the compound to 66%.

The NAC dose was expressed in mg per kilogram of body weight by day, and the respective adjustments were made throughout the document. In the discussion section of the manuscript (Lines 322-325), it has been established that the NAC use in birds has been developed previously, and the protective properties have been shown; This confirms what is stated in the introduction section of the manuscript (Lines 61- 66).

  1. -Section 3: How do the authors explain the differences observed in hepatic GSH, hepatic GGT, ALT, and relative liver and kidney weights when NAC or EQ is administered alone compared with the control diet without AFs (0.0 mg/kg)? These differences should be addressed in the discussions section. In addition, is there statistical significance for the comparisons between NAC alone vs. control diet and EQ alone vs. control diet?

  1. -Section 5.1: The number of laying hens per cage should be specified.

In lines 493-495 it stated that at 14 weeks of age, one bird or experimental unit was divided (one per cage) into 12 experimental treatment groups,

  1. -Section 5.2: What were the concentrations of ochratoxin (OTA), fumonisins (FBs), zearalenone (ZEA), and deoxynivalenol (DON) determined using competitive ELISA kits?

As in other studies on natural mycotoxin contamination in hen feed, low concentrations of ZEN, OTA, DON, and FB were found. The presence of FB, DON, and OTA was below the minimum detection limits, while ZEN concentration was only detected occasionally during weeks 14 and 46 of the study (63.1 ± 24.4 μg/kg). The concentration of ZEN in feed that could cause a decrease in bird performance is >50 mg/kg; however, in this study, the observed ZEN concentration was several times lower than that concentration, suggesting that the health changes detected in this study were not caused by ZEN or other mycotoxins.

  1. Details regarding the characterization of the diet should also be provided, particularly the physicochemical parameters. These could be briefly described in the methodology section or presented in a separate subsection of the results. These data can help other researchers in interpreting these results or in reproducing the experiment in future results.

In lines 477-484, it is stated that the diet for laying hens was designed to satisfy the nutritional requirements of this phase.

  1. -Section 5.3: Were the kidney and liver samples for GSH, GST, GGT, and total protein analysis homogenized immediately after collection? Were the tissue samples or homogenates stored at -20 °C or -80 °C prior to analysis? The description of the sample processing for these parameters is unclear and should be specified.

The methodological processes were reviewed and rewritten (Line 539-553).

  1. -Section 5.5: It may be clearer if the section about experimental design is placed at the beginning of the methodology, so that the presentation follows the actual sequence of the study.

The experimental design section has been relocated to the beginning of the methodology in the revised version of the manuscript to improve clarity and better reflect the actual sequence of the study.

  1. -Section 5.6: Whether data from technical repetitions were averaged?

The procedure for averaging the assessments made by the pathologists who participated in evaluating the lesions was explained.

  1. -Section 5.6: What was the power of used statistical tests?

Response: In the statistical analysis section of the manuscript, the procedure for calculating the statistical power of the tests for continuous variables and frequency of lesions, the value achieved, and the bibliographic reference corresponding to this procedure have been added.

Minor suggestions:

  1. -Avoid using the long dash (—) in phrases such as “A significant reduction in biochemical markers of aflatoxicosis—such as ALT and AST—was observed”. This style is often characteristic of AI generated text.

Response: Long dashes (—) were removed from sentences in all sections of the manuscript, and the texts were reworded.

  1. -Section 2: Consider shortening the explanations in the results section, since similar points are already covered in the discussions section

The entire manuscript was reviewed again, and several sections were improved, especially the conclusions. All changes are highlighted in yellow.

Round 2

Reviewer 1 Report

Comments and Suggestions for Authors

Dear Authors,

Thank you for the substantial work you have invested in revising the manuscript. We particularly appreciate: (i) inclusion of validation parameters for the HPLC aflatoxin assay, (ii) clarification of methods and statistics (experimental unit; post‑hoc power), (iii) improved presentation of histology (higher magnification and clear scale bars), (iv) the dosing rationale for EQ/NAC with conversion to mg/kg BW/day, and (v) the explicit acknowledgement of the lack of residue data as a key limitation. These changes noticeably improve the clarity and rigor of the paper.

At the same time, several methodological comments could not be fully implemented at this stage—understandably so, as they would require additional experiments or re‑analysis. In particular, we note the absence of multi‑mycotoxin LC–MS/MS confirmation, special stains to distinguish hepatic steatosis from glycogen/hydropic change, more appropriate modeling for ordinal histology outcomes, recorded actual feed intake, and compliance biomarkers for NAC. We recognize these are difficult or impossible to address within the current revision cycle.

We ask only that these limitations be consistently and visibly flagged in the Methods/Discussion/Conclusions and that interpretations remain calibrated to the available evidence.

Thank you again for your thoughtful and constructive revisions.

Author Response

REVIEWER COMMENTS

Several methodological comments could not be fully implemented at this stage—understandably so, as they would require additional experiments or re‑analysis. In particular, we note the absence of multi‑mycotoxin LC–MS/MS confirmation, special stains to distinguish hepatic steatosis from glycogen/hydropic change, more appropriate modeling for ordinal histology outcomes, recorded actual feed intake, and compliance biomarkers for NAC. We recognize these are difficult or impossible to address within the current revision cycle. We ask only that these limitations be consistently and visibly flagged in the Methods/Discussion/Conclusions and that interpretations remain calibrated to the available evidence.

RESPONSE TO REVIEWER COMMENTS

The authors sincerely thank you again for your suggestions and comments, which significantly improved the manuscript. We also appreciate your patience in reading and trying to understand the text in detail, as well as your ability to offer helpful comments. Below, we inform you how we have addressed your observations and comments.

  1. Multi-mycotoxin LC–MS/MS Confirmation

The explanation of the methods used to quantify mycotoxins in this study was expanded in the discussion section of this manuscript (lines 312-324). Liquid chromatography-tandem mass spectrometry is widely used as the reference method for routine mycotoxin determination. The main advantages of this method are its low detection and quantification limits, as well as its applicability to different feed varieties, as it is a simple, fast method suitable for the simultaneous determination of multiple mycotoxins. Although in this study, the concentration of aflatoxins was determined by HPLC, the absence of other mycotoxins was verified by an indirect enzyme-linked immunosorbent assay, following extraction, purification in immunoaffinity columns, and comparison with the results of purified mycotoxin standards. This ELISA procedure for the determination of mycotoxins is acceptable to several authors, who agree that there is a very high correlation (R2>90%) between the concentrations of mycotoxins in feed and animal products, estimated comparatively between ELISA and LC-MS/MS. Therefore, it can be assumed that both techniques are satisfactory in terms of meeting feed safety control basic requirements.

  1. Special Stains

The explanation of the methods used to detect fatty liver degeneration in this study was expanded in the discussion section of this manuscript (lines 439-450). The specific stains, Oil Red O or PAS with diastase, are highly efficient and enable the distinction between hepatic steatosis and lipid accumulation, glycogen vacuolization, or hydropic changes; however, in this study, hematoxylin and eosin staining was used to detect fatty liver degeneration; these tissue changes were detected with sufficient accuracy to identify them as hepatic steatosis. This staining has also been described in other studies with birds and mammals and has been linked to alterations in lipid metabolism enzymes. H&E staining remains a widely accepted method for identifying both microvesicular and macrovesicular hepatic steatosis. H&E staining has also been reported as a method for identifying lipid accumulation in the liver, provided that specific morphological criteria are considered. These criteria include the presence of intracytoplasmic vacuoles, peripheral displacement of the nucleus, the appearance of the vacuoles, and the characteristic zonal distribution.

  1. Modeling for Histology Outcomes

In tables 3 and 4, also in the materials and methods section of the manuscript, a model for histology outcomes (lines 615-622) was included. Histological analysis was performed independently (blinded) by two pathologists, and a third pathologist participated in discrepancies; for each specimen, only the result that matched between the two pathologists was recorded. The tissues were examined for a homogeneous distribution of lesions in the hepatic lobes or renal regions, so each specimen was reported as a single unit. Microscopic alterations were analyzed as categorical data (1 = lesion, 0 = no lesion detected) by estimating the proportion of lesions identified in 90 readings of 15 necropsies for each treatment; Bonferroni's post hoc multiple comparison procedure was used.

  1. Actual Feed Intake

The procedure for estimating feed consumption was specified in the materials and methods section of the manuscript (Lines 515-518). At 12 weeks of age, each hen was housed in an individual cage with a separate feeder, and the daily feed consumption was recorded for each bird. All animals were closely observed until 72 weeks of exposure. Body weight and egg production were recorded, and feed efficiency was calculated.

  1. Biomarkers for NAC

The explanation of evaluating the biomarkers for NAC in this study was expanded in the discussion section of this manuscript (lines 336-346). Diverse studies have evaluated the main biomarkers derived from the therapeutic administration of NAC to validate its bioavailability and efficacy, especially cysteine, glutathione, and the enzymes involved in its binding to xenobiotic compounds. In this study, the concentration of GSH in liver and kidney tissues was measured, as well as the enzymatic activity of glutathione S-transferases; however, no distinction was made between compounds derived from the ingestion and metabolism of NAC and the de novo synthesis of GSH or the enzymatic induction of GST, so their concentration or activity cannot be considered a direct biomarker of the chemoprotective efficacy of NAC treatment. Nevertheless, the biological effects of NAC treatment were evaluated through functional, biochemical, and histopathological parameters in birds intoxicated with AFs.

Thank you again for your thoughtful and constructive revisions.

Reviewer 2 Report

Comments and Suggestions for Authors

The article has been improved, but one point still requires attention. The authors did not respond, or may have accidentally overlooked, the third comment: “-Section 3: How do the authors explain the differences observed in hepatic GSH, hepatic GGT, ALT, and relative liver and kidney weights when NAC or EQ is administered alone compared with the control diet without AFs (0.0 mg/kg)? These differences should be addressed in the discussions section. In addition, is there statistical significance for the comparisons between NAC alone vs. control diet and EQ alone vs. control diet?”. I ask the authors to clarify.

Author Response

REVIEWER COMMENTS

The article has been improved, but one point still requires attention. The authors did not respond, or may have accidentally overlooked, the third comment: “-Section 3: How do the authors explain the differences observed in hepatic GSH, hepatic GGT, ALT, and relative liver and kidney weights when NAC or EQ is administered alone compared with the control diet without AFs (0.0 mg/kg)? These differences should be addressed in the discussions section. In addition, is there statistical significance for the comparisons between NAC alone vs. control diet and EQ alone vs. control diet?”. I ask the authors to clarify.

RESPONSE TO REVIEWER COMMENTS

The authors sincerely thank you again for your suggestions and comments, which significantly improved the manuscript. We also appreciate your patience in reading and trying to understand the text in detail, as well as your ability to offer helpful comments. Below, we inform you how we have addressed your observations and comments.

Figures 1 to 6 in the results section of the manuscript show the effects of the combination of NAC and EQ at the four levels of AFs; however, a comparison of the NAC and EQ treatments without aflatoxins (T5, T9, Table 5) is not included because the results are like the control group without aflatoxins or chemoprotectors (T1). For this reason, in the results and conclusions sections of the manuscript (Lines 92-94, 114-117, 234-236, 476-478), a description was included of the absence of changes in production parameters, plasma or tissue biochemistry, or macroscopic or histological alterations in the liver or kidneys following the ingestion of NAC or EQ, compared to hens that did not consume aflatoxins. In addition, the explanation of evaluating the biomarkers for NAC in this study was expanded in the discussion section of this manuscript (lines 336-346). Diverse studies have evaluated the main biomarkers derived from the therapeutic administration of NAC to validate its bioavailability and efficacy, especially cysteine, glutathione, and the enzymes involved in its binding to xenobiotic compounds. In this study, the concentration of GSH in liver and kidney tissues was measured, as well as the enzymatic activity of glutathione S-transferases; however, no distinction was made between compounds derived from the ingestion and metabolism of NAC and the de novo synthesis of GSH or the enzymatic induction of GST, so their concentration or activity cannot be considered a direct biomarker of the chemoprotective efficacy of NAC treatment. Nevertheless, the biological effects of NAC treatment were evaluated through functional, biochemical, and histopathological parameters in birds intoxicated with AFs.

Thank you again for your thoughtful and constructive revisions.